# Embedding electronic patient-reported outcome measures into routine care for patients with stage III MELanoma (ePROMs-MEL): protocol for a prospective, longitudinal, mixed-methods pilot study

Kathy Dempsey ![ORCID] ,[1,2] Robyn Saw,[2,3] Iris Bartula,[3] Serigne Lo,[3] Craig Lawn,[3] Thomas Pennington,[3] Andrew Spillane,[2,3] Frances Boyle,[2,3] Skye Dong,[3] Mbathio Dieng,[1,2] Donna Milne,[4] Linda Seaman,[3] Dina Saks,[3] Julia Lai-Kwon,[3] Jake Robert Thompson ![ORCID] ,[3] Rachael Morton[1,2,3]

For numbered affiliations see end of article.

**Correspondence to**
Dr Kathy Dempsey;
kathy.dempsey@sydney.edu.au

## ABSTRACT

**Introduction** The benefits of patient-reported feedback, using questionnaires that allow patients to report how they feel and function without any interpretation from healthcare professionals, are well established. However, patient-reported outcomes measures (PROMs) are not routinely collected in patients with melanoma in Australia. The aim of this study is to evaluate the feasibility and acceptability of implementing electronic PROMs (ePROMs) into routine care from the perspectives of patients with stage III melanoma and their treating clinical team.

**Methods and analysis** A minimum of 50 patients and 5 clinicians will be recruited to this prospective, longitudinal pilot study (ePROMs-MELanoma). The study uses a mixed-methods approach (quantitative PROMs questionnaires and end-of-study surveys with qualitative interviews) and commenced in May 2021 in surgical and medical melanoma clinics at two sites in metropolitan Sydney, Australia. The primary outcomes are measures of feasibility and acceptability, comprising descriptive questionnaire completion statistics, and proportion of patients who reported that these PROMs were easy to complete and measured items they considered important. Clinician and clinic staff views will be canvassed on the appropriateness of these PROMs for their patients, change in referral practice and uptake and incorporation into routine practice. Secondary aims include measurement of improvements in patients' emotional and physical health and well-being, and utility of real-time data capture and clinician feedback. All participants will complete the Distress Thermometer and EQ-5D-5L questionnaires in the clinic using a tablet computer at baseline and two to three subsequent follow-up appointments. Participants who report a score of 4 or higher on the Distress Thermometer will be triaged to complete an additional three questionnaires: the QLQ-C30, Depression, Anxiety and Stress Scale and Melanoma Concerns Questionnaire-28. Results will be generated in real time; patients with psychosocial distress or poor quality of life will discuss

## STRENGTHS AND LIMITATIONS OF THIS STUDY

⇒ This prospective, longitudinal cohort intervention pilot study, conducted at two sites, examines the perspectives of patients, clinicians and clinic staff.
⇒ A stakeholder reference panel, with consumer representation, was formed in the planning stages of this study and is actively involved in oversight of the study.
⇒ The study uses a triage system of distress screening to minimise survey burden for those who are not distressed and maximise information collection from those who are.
⇒ Summary reports from the completed questionnaires, including the first use of the Melanoma Concerns Questionnaire-28, a bespoke melanoma patient-reported outcome measures questionnaire, are immediately forwarded to the treating clinician in real time, so they can be discussed during the patient's consultation.
⇒ Limitations include the absence of a control group and an unavoidable reduction in the participant enrolment target due to the impact of COVID-19 restrictions on recruitment.

possible referral to appropriate allied health services with their clinician. Thematic analysis of interviews will be conducted.

**Ethics and dissemination** Ethics approval obtained from St Vincent's Hospital Human Research Ethics Committee on 19 September 2019 (2019/ETH10558), with amendments approved on 8 June 2022. Patient consent is obtained electronically prior to questionnaire commencement. Dissemination strategies will include publication in peer-reviewed journals and presentation at international conferences, tailored presentations for clinical societies and government bodies, organisational reporting through multidisciplinary meetings and research symposia

for local clinicians and clinic staff, and more informal, lay reports and presentations for consumer melanoma representative bodies and patient participants and their families.

**Trial registration number** ACTRN12620001149954.

## INTRODUCTION

Patient-reported outcome measures (PROMs) are questionnaires that allow patients to report how they feel and function without any interpretation from healthcare professionals.[1] PROMs can be used in the management of individual patients to screen for a particular condition (such as depression) and/or to monitor patient response to an appropriate intervention (such as psychological therapy).[1] Previous research has shown that the use of PROMs can improve clinician–patient communication. One study of women with gynaecological cancers found a PROMs questionnaire helped with discussion of symptoms not typically addressed for nearly all participants (95%, n=40) and improved patient care for 97% (n=41) of participants.[2] Other studies have found PROMs can facilitate clinician awareness and improve patient management,[3 4] contribute to prognostic information,[3] potentially prolong survival outcomes in patients with advanced cancer,[5] as well as allowing patients to play a greater role in the clinical decision making process.[1] Currently, PROMs are not routinely collected as part of melanoma treatment in Australia.

It has long been recognised that many cancer specialists are not good at identifying cues for depression and psychological distress.[6 7] Routine screening for distress is now recommended by theUS National Comprehensive Cancer Network[8] and the International Psycho-Oncology Society and 68 affiliated organisations have set a standard of care involving monitoring distress as the '6th vital sign'.[9] Australian guidelines support the use of electronic PROMs (ePROMs) for the timely detection of psychosocial distress, but the authors note that results of trials evaluating the efficacy of distress screening have been mixed.[9] Effective trials used strategies that minimised staff burden and ensured effective follow-up.[9]

Previous research has demonstrated increased anxiety, depression and lower quality of life in melanoma patients.[10–12] An Australian review of psychosocial support estimated 30% of all melanoma patients report levels of psychological distress indicative of the need for clinical intervention.[13] Stage III melanoma patients usually have more morbid surgery, including sentinel node biopsy or lymph node dissection, which impacts significantly on their quality of life. When patients with stage III melanoma attend follow-up appointments with their treating physician, the focus is primarily on detection and treatment of recurrence of disease. There is minimal focus on assessing the patients' quality of life or psychological health status, despite recommendations from The Cancer Institute New South Wales (NSW)[14] and the Australian Commission on Safety and Quality in Health Care.[15] An audit of Australian cancer services reported that 'evidence

from clinical settings suggests timely identification of distress is only effective in improving medical management and patient well-being when paired with structures supportive care referrals'.[16]

A 2022 report on melanoma in Australia[17] noted the strong evidence that:

► Access to emotional support services improves the well-being of people with cancer.
► The opportunity to discuss feelings with a member of the treatment team or counsellor decreases psychosocial distress.
► Psychoeducational programs decrease anxiety and depression.

This landmark report concluded that increased screening for supportive care services should be developed as a quality indicator of melanoma clinical care.[17]

While there has been extensive research on estimating the extent of psychosocial distress in a vast range of clinical settings, subsequent translation of that knowledge into programmes and services currently lags behind gains in the medical treatment of advanced melanoma.[18] This gap between obtaining results from PROMs and using this information as the first step in a process to actually improve patient outcomes is an overdue and urgent area of research.

By including the collection and evaluation of ePROMs data into the routine care for patients with stage III melanoma, the treating clinician will be able to identify patients who are experiencing deterioration in quality of life or psychological distress. Once identified, these patients may then be referred to the appropriate allied health professionals (eg, psychologist, specialist nurse, physiotherapist, occupational therapist and social worker) for appropriate support.

The primary aim of this pilot study is to evaluate the feasibility and acceptability of implementing ePROMs into routine care from the perspectives of both stage III melanoma patients and their treating clinical team (ie, surgeons, dermatologists, medical oncologists, psychologists and melanoma nurses). If the pilot results demonstrate technical, administrative and logistical feasibility, as well as patient, clinician and financial acceptability, this pilot will be expanded to all melanoma clinics within Melanoma Institute Australia (MIA). This expansion will provide an opportunity to collect and analyse a greater volume of data and to potentially serve as a model for wider roll-out of ePROMs across the public and private healthcare sectors in NSW and other states and territories of Australia.

## METHODS AND ANALYSIS
### Patient and public involvement

A stakeholder reference panel was formed in the planning stages of this pilot study. Members of this panel included experts in the field of melanoma (various clinical and non-clinical specialties), experts in patient-reported measures research, and a small group of representatives/

advocates for patients with current and previously treated melanoma. Consumer involvement is considered vital to ensure that the needs and preferences of patients were incorporated alongside the knowledge and background of treating clinicians and researchers.

This stakeholder representative panel was responsible for drafting a detailed project proposal to be funded as part of the Australian National Health and Medical Research Council Centre for Research Excellence in Melanoma. Panel tasks included: selection of the most appropriate PROMs to be included in this study, with consideration of minimising patient burden while maximising data usefulness; the content of semistructured qualitative interview schedules (for patients and clinicians); development of all patient information documents, including the Participant Information Statement and Consent Forms (PISCFs), an Expression of Interest (EoI) form and the 'About you' demographic patient information collection form; and clinician education resources including the development of short videos about the project and advice on discussion structure, in both electronic and written formats.

In addition to the documents, the manner of recruitment was widely discussed at these meetings, as was the creation of a comprehensive REDCap database that would receive results directly from the i-Pad custom-designed software while ensuring patient confidentiality. Consumer views also strongly influenced the plan for dissemination of results to patients, clinicians and their general practitioners.

### Study design and setting
ePROMs-MELanoma is a prospective, longitudinal intervention study incorporating mixed methods (quantitative surveys and qualitative interviews). Although there is no control group, the design incorporates before-and-after study design elements allowing for comparisons of psychosocial health and quality of life in individuals with stage III melanoma before and after the introduction of ePROMs. This study will be conducted at surgical and medical melanoma clinics at MIA and at Royal Prince Alfred Hospital (RPAH), Sydney, NSW, Australia.

### Eligibility criteria and sample size
Patients are eligible to participate if they are aged 18 or over, have received a diagnosis of Stage III melanoma at least 1 month earlier, are under the care of melanoma clinicians at MIA or RPAH, and have sufficient English and cognitive ability to comprehend study materials, provide informed consent and participate in the study. Clinicians, including surgical oncologists, medical oncologists, dermatologists and melanoma general practitioners, are eligible to participate if they are currently treating and managing patients with stage III melanoma at MIA or RPAH.

This research project has experienced substantial delays to recruitment due to research staff being unable to enter clinic sites during COVID-19. In April 2022, the decision was made to reduce the initial recruitment target of 100–50 patients, so that the trial may be completed before external funding expires (the minimum number of clinician participants is five). The target reduction is justified on the basis that this project is a non-randomised feasibility pilot study. Assuming 30% of stage III melanoma patients have levels of distress that indicate the need for clinical intervention,[13] we would expect to identify approximately 17 patients within our sample who will be triaged for discussion of referrals to support services. This number is considered sufficient to inform a decision on whether or not to proceed to a larger-scale trial with additional sites.

### Recruitment
Recruitment has been intermittent due to delays caused by COVID-19 (April–September 2020 and June 2021–March 2022). Recruitment commenced at MIA on 28 May 2021 and at RPAH on 5 July 2021. Recruitment was extended to medical oncology clinics at MIA in March 2022 to bolster recruitment numbers. Thirty-six patients and seven clinicians have been recruited since the study commenced in May 2021, with the study expected to be completed by June 2023.

### Patient recruitment process
A study clinician will review a list of potentially eligible patients (with stage III melanoma) obtained from the MIA Melanoma Research Database to check clinical eligibility (ie, are still under the care of MIA clinicians at either site or have not progressed to stage 4 melanoma). Updated lists will then be sent to each of the treating clinicians who will be able to remove patients they feel are not appropriate to invite to participate in the study for other reasons. The clinical nurse consultant or clinic secretary will then send the approved list of eligible patients to the project officer.

These patients will be initially contacted by the project officer prior to their scheduled clinical review appointments. The project officer will send them three documents along with a covering letter (either by email or post; posted documents will contain a stamped self-addressed envelope). The email/letter will briefly introduce the study, ask patients to read the invitation to participate and the participant information statement (PIS), and to complete and return an EoI form, which provides patients with three options to select from: willing, potentially willing or not willing to participate. If patients do not respond to the request to return the EoI, the project officer will telephone them once to check whether they have received it.

The project officer will follow up those patients who have indicated their willingness or potential willingness to participate, answer any questions and concerns they may have about the study and ask them to arrive 30 min earlier for their scheduled appointment to allow time to complete the questionnaires. The PIS contains contact numbers of the academic and clinical leads for this project so patients may contact them prior to making

their decision to participate. When patients arrive at the clinic for a regular appointment, they will again be given the opportunity to discuss any questions they may have regarding their participation with the project officer who will greet them, obtain their electronic consent on the iPad and show them how to complete the questionnaires on the iPad.

Patients who were recruited with stage III melanoma but progress to stage IV disease during the study can choose whether to continue in the study or not. Any continuing stage IV patients will be asked to complete an interview at the end of the study to provide an assessment of how helpful they found this project to be considering their changing treatment and needs.

### Clinician recruitment process

Clinical staff will be informed about the project through education sessions at clinical and multidisciplinary team (MDT) meetings and be given a written invitation to participate. Interested clinicians will be provided with an electronic PISCF to read. Either hard copies of signed consent or scanned electronic versions will be acceptable and collated by the project officer prior to the clinician seeing study patients. All clinicians who were approached to participate consented to the study and reviewed the study questionnaires with the patients.

### Interventions

The study questionnaires will be repeated at subsequent regularly scheduled follow-up appointments with their clinicians, with a minimum of 2 months between patient completion of questionnaires. Questionnaires will be administered at baseline, with a minimum of two and a maximum of three, follow-up time points. iPads were chosen as the delivery platform because they are portable, have a user-friendly interface, are capable of hosting software that can automatically generate reports and associated documents and are a secure format for data management as they are encrypted by default. All questionnaires used in this study have been validated in previous studies.

At each visit, all patients will be required to complete two short questionnaires that are used to assess their psychosocial distress and quality of life: (1) the National Comprehensive Cancer Network's Distress Thermometer and Problem List for Patients[19] and (2) EuroQoL's EQ-5D-5L (five dimensions, five levels) and Visual Analogue Scale.[20] Patients who score above the clinical cut-off score on the Distress Thermometer (4 or above) will be triaged to complete an additional three questionnaires: the Melanoma Concerns Questionnaire (MCQ-28),[21] the Depression, Anxiety and Stress Scale (DASS-21)[22] and the European Organisation for Research and Treatment of Cancer (EORTC)'s Quality of Life Questionnaire - Cancer (QLQ-C30).[23] These additional questionnaires will provide more detailed information about potential areas of support needs. Completion time for all five questionnaires is expected to be less than 30 min. All questionnaires are provided in online supplemental appendix 1.

Results from the completed questionnaires will be automatically generated, emailed to the clinic secretary and printed for the treating clinician in real time, so the results can be discussed during their consultation,

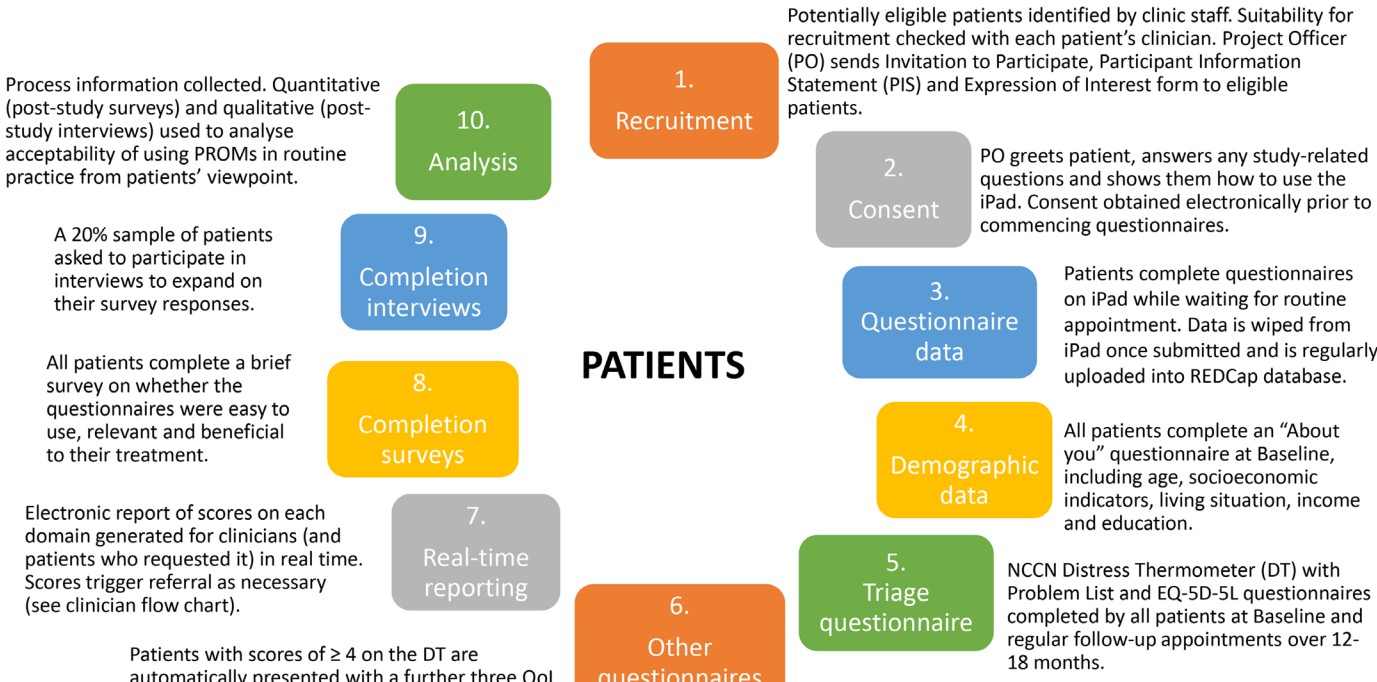

**Figure 1** Patient's perspective.

along with medical concerns. Patients who request a copy of their results (by selecting the appropriate box in the 'About You' survey) will receive a summary of their scores by email, prior to their imminent appointment. Participants who score 3 or above on any of the five domains covered in the EQ-5D-5L questionnaire[20] (mobility, self-care, usual activities, pain/discomfort and anxiety/depression) may require clinical intervention (3=moderate, 4=severe, 5=extreme limitations). There are no melanoma-specific cut-off scores indicating the need for clinical intervention, for the QoL questionnaires MCQ-28[21] and EORTC QLQ-C30.[23] Changes in scores for these two questionnaires will be recorded at each time point, allowing for manual inspection of changes over time. A deterioration in EORTC QLQ-C30 scores of 10 points or more between visits is commonly regarded as clinically significant.[24 25] Recommendations and referrals for additional support services will be based on the questionnaire scores and clinical experience of the clinicians and be made in collaboration with the patient. Further information on reporting processes is available in online supplemental appendix 2.

After all rounds of patient questionnaires (baseline plus 2–3 follow-up rounds per patient) have been completed, all patients, clinicians and clinic staff involved in the project will be asked to complete an electronic survey providing quantitative and open-ended assessments of their experiences with the PROMs questionnaires. In addition, approximately 20% of patients (10 patients), all study clinicians and clinic staff with major involvement in the study will be invited by the research team to participate in a short (10 min) confidential interview to expand on their survey responses. Patients will be purposively sampled to maximise response variation, based on a range of individual views and experiences as identified from their survey results. Patients who complete an interview will be offered a US$30 gift card to thank them for their time, while clinicians and clinic staff who participate in an interview will not be reimbursed. Figure 1 provides a study flow chart from the patient's perspective and figure 2 provides the study overview from the clinician's perspective.

## Outcomes

The primary intended outcome of this study is to determine whether completion of PROMs electronically are considered by both patients, clinicians and clinic staff as feasible and acceptable when incorporated into routine clinical care. Secondary outcomes include assessment of whether these measures help identify people who need and would benefit from further support/referral; the production of a framework for the data systems required to enable electronic data capture and feedback to clinicians, useful in many other applications and clinical specialties; and the utility of clinician training resources. The primary outcome has three components.

### Patient perceptions

Patient ratings of feasibility and acceptability will be assessed in several ways. The first is by the proportion of eligible patients approached who completed baseline and at least one set of follow-up PROMs data. If 60% of patients meet this requirement, the study will be deemed feasible and acceptable overall.

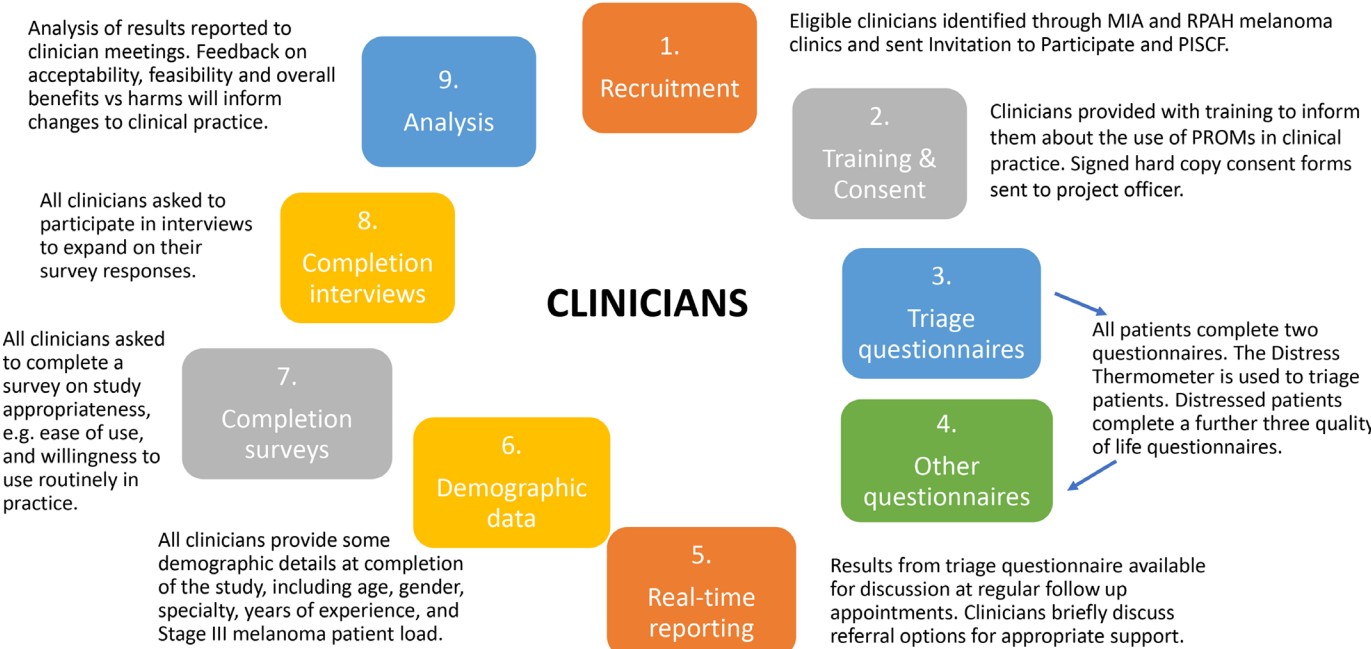

**Figure 2** Clinician's perspective.

Data on individual patient measures of feasibility and acceptability will also be collected. In the first set of two triage questionnaires, all questions are mandatory, but with the second set of three questionnaires, patients are not required to answer all questions. Metadata from the patients who are triaged to complete the second group of questionnaires will be collected to provide an estimate of the mean time taken to complete the PROMs questionnaires and the amount of missing data (providing a second measure of patient acceptability for those who completed all five questionnaires). If patients take more than 30 min to complete the five questionnaires, they will be considered as not having met this acceptability criteria. Additional measures will include changes in baseline and follow-up scores and the proportion of patients who: 'agree or strongly agree' that the PROMs were easy to complete and that the questions they completed measured issues they considered important to their QoL (online supplemental appendix 3a). Semistructured interviews

with patients will also assess feasibility and acceptability through the following main question themes: enjoyment of completing the PROMs; changes in patient–clinician communication; changes in their management; and improvements or challenges they found in the processes.

### Clinician perceptions

It is also important to assess the feasibility and acceptability of this project from the clinicians' perspective. Measures include whether they found the PROMs data provided to them to be useful, timely, easy to understand and appropriate for their patient population. Clinicians will be approached to expand on their survey responses (online supplemental appendix 3b) in a semistructured interview. The key themes to address are related specifically to the PROMs instruments and the information generated for the clinician; changes in patient management; improvements in patient–clinician communication; and any key challenges or improvements that they can identify regarding the processes.

### Clinic staff perceptions

Clinic staff including clinical nurse consultants and medical secretaries will also be asked to assess the feasibility and acceptability of incorporating QoL questionnaires into their clinic's routine practice (online supplemental appendix 3c). If they consider the additional workload or clinic disruption to be unacceptable or burdensome, then this will have a major impact on the feasibility assessment of continuing this practice once the pilot study has been completed.

### Secondary outcomes

Research and clinical psychologists at MIA will use identified unmet needs from this study to inform future research into the development of support programmes at MIA. The extent to which the routine use of PROMs in clinical practice can improve patients' emotional and physical problems will be assessed through changes in referral rates, referral uptake rates and patients' QoL scores over time. Patient and clinician interviews will allow for more extensive evaluation of this aspect of the study.

The utility of this framework for the data systems required to enable electronic data capture and feedback to clinicians will be assessed in patient, clinician and clinic staff surveys and interviews as well as assessment by the project team. End-of-study surveys will highlight the perceived ease of use, usefulness, relevance and importance of ePROMs to each group (online supplemental appendix 3). Data will also be collected on time taken to complete questionnaires and on the number of times the questionnaires were completed (out of a maximum of 3–4 time points). In addition, the number of questions answered in the second set of three questionnaires (all questions in the first set of two questionnaires are mandatory) will be recorded.

**Table 1** Study interventions and assessments timeline

| | Baseline | | | End of study |
|---|---|---|---|---|
| Time points* | T0 | T1 | T2 | T3 |
| Enrolment | | | | |
| Eligibility screen | X | | | |
| Informed consent | X | | | |
| Recruitment | X | | | |
| Interventions | | | | |
| Baseline Patient Questionnaires | X | | | |
| Baseline patient demographic info | X | | | |
| Follow-up Patient Questionnaires | | X | X | X |
| Clinician Completion Surveys | | | | X |
| Patient Completion Surveys | | | | X |
| Clinician completion interviews | | | | X |
| Patient completion interviews | | | | X |
| Clinician referral data | | | | X |
| Patient referral uptake data | | | | X |
| Patient change over time (T0–T5) | | | | X |
| Assessments | | | | |
| Clinician acceptance | | | | X |
| Patient acceptance | | | | X |
| Clinic staff acceptance | | | | X |
| Referral rate | | | | X |
| Referral uptake rate | | | | X |

Table 1 adapted from the template designed by the SPIRIT Group © 2013.[29]
*For each time point, a window of ±2 months is acceptable due to variation in routine clinical appointments.
SPIRIT, Standard Protocol Items: Recommendations for Interventional Trials.

**Table 2** Assessment of implementation outcomes[26]

| Implementation outcome | Definitions | Examples of assessment measures |
| --- | --- | --- |
| Appropriateness | 'The perceived fit, relevance or compatibility of the innovation or evidence based practice for a given practice setting, provider, or consumer; and/or perceived fit of the innovation to address a particular issue or problem.' | Surveys and interview questions with participating clinicians and clinic staff. |
| Costs | 'Tthe cost impact of an implementation effort.' (not restricted to financial costs). | Costs to patients: estimates of time (need to arrive earlier for appointments) and inconvenience (eg, heavier traffic; questionnaire fatigue). Costs to clinicians: estimates of time to incorporate patient responses into discussion; opportunity costs. Costs to clinic staff: estimates of time required to provide recruitment information to researchers, to print out questionnaire responses, to scan these for the patients' records. Calculation of financial costs of research staff required to manage the administration and analysis of PROMs and the reporting of study outcomes. |
| Fidelity | 'The degree to which an intervention was implemented as it was prescribed in the original protocol or as it was intended by the programme developers.' | Documentation of variations from original protocol requiring revisions and further ethics approval. |
| Adoption | 'The intention, initial decision, or action to try or employ an innovation or evidence-based practice. Adoption also may be referred to as 'uptake'.' | Assessment of the willingness of all participants to engage with project. Barriers and enablers of uptake will be explored in surveys and interviews. |
| Penetration | 'The integration of a practice within a service setting and its subsystems.' | Documentation of additional resources required to manage the project in the clinical setting as well as assessment of the impact of barriers to integration for example, lack of clinical champions. |
| Sustainability | 'The extent to which a newly implemented treatment is maintained or institutionalised within a service setting's ongoing, stable operations.' | Long-term assessment of the continued use of these PROMs in study clinics and any extension to other clinics. |

PROMs, patient-reported outcome measures.

All participating clinicians are experienced in treating stage III melanoma patients and will be asked to review resources prepared for this study. These include three short videos about PROMs, referral pathways and the project details and support services referral information on the MIA intranet. The aim of these resources is to help clinicians familiarise themselves with the objectives of the study and the use of PROMs in an oncology setting. Online supplemental appendix 4 contains a list of study investigators. Table 1 provides a timeline of expected outcomes.

The overall success of the project will be assessed using the Proctor *et al* framework for implementation research,[26] which considers not only the intervention outcomes (service and client) but also the implementation outcomes. Table 2 defines the implementation outcomes (excluding acceptability and feasibility which are described throughout the protocol) and provides examples of how they will be measured in the context of this study.

## Data analysis and statistical methods
### Quantitative analysis
Participant demographics and characteristics will be summarised for all participants using means and SD. In addition, the consent, participation and completion rates will be analysed. Consent will be defined as agreeing to participate in the study and returning a signed consent form; participation, as completing some of the components of the study; and completion, as completing all components of the study. Specifically, the proportion of eligible participants who completed the project tasks will be assessed to determine whether the inclusion of ePROMs are acceptable for patients undergoing follow-up treatment for stage III melanoma. Descriptive statistics will be reported for QoL and other questionnaire domain scores at baseline and follow-up time points. Referral and referral uptake rates will also be calculated. The small sample size of 50 for this pilot study precludes any subgroup analysis. Linear mixed effects modelling will be used to account for repeated measures and differential

time points among participants. This is similar to analysis of longitudinal data from a clinical trial where measurements at follow-up time points are missing.[27]

### Qualitative analysis

This study takes an inductive approach to data analysis, with no predetermined hypotheses about what we expect to find. Results from semistructured patient interviews conducted at the conclusion of the project will be reported to assess patient views on the usability, benefits and limitations of the study questionnaires they completed, whether questionnaire results were discussed within the subsequent clinical consultation, and if so whether this led to any change in management. Similarly, results from end of study clinician interviews will report their views on the feasibility, acceptability, benefits, limitations and usefulness of including ePROMs into the routine care for patients within their practice. Electronic qualitative analysis software (NVivo) will be used to assist in the creation of a framework of key themes identified from the interview data that will then inform the development of an iterative thematic analysis process.[28]

### Ethics and dissemination
#### Ethics and consent

Original ethics approval was obtained from St Vincent's Hospital Human Research Ethics Committee (HREC) on 19 September 2019 (2019/ETH10558). This approval included site authorisation for the following two project sites:

▶ Sydney Melanoma & Surgical Oncology, MIA, Poche Centre.
▶ Sydney Melanoma & Surgical Oncology, RPAH, Camperdown.

Amendments to the original protocol were accepted by St Vincent's HREC on 8 June 2022.

Consent is obtained electronically on the iPad from all patients prior to commencing the questionnaires, while clinical staff signed written consent forms. All participants who complete end-of-study surveys or interviews will sign additional written consent forms. There are no foreseeable additional harms, risks or costs associated with taking part in this study that are not outlined in the PIS. It is possible some patients may experience short-term distress while answering the questionnaires. If a patient does not wish to answer a question, they may skip it and go to the next question (if they are completing the second set of questionnaires), or immediately stop completing the questionnaires (if they are completing the first set of questionnaires, where all questions are mandatory). A melanoma clinical nurse consultant and/or treating clinician will be available to talk with a distressed patient if their distress continues once they have stopped completing the questionnaire. Patients are free to withdraw from the study at any time. Clinicians may experience frustration with having to review patient summary scores from the questionnaires if they are particularly busy.

### Dissemination of trial results

Study results will be disseminated to participants via a short, lay summary of overall study findings at the end of the study. Patients can also opt in to receive their personal questionnaire results at each scheduled visit over the study time period. Overall study results will be communicated to participating clinicians and local healthcare professionals through MDT meetings and to wider clinician and policy audiences through presentations to the national surgical oncology, medical oncology and nursing representative societies, as well as government bodies. In addition to the standard fora of publication in peer-reviewed journals, seminar/webinar presentations and presentations at scientific conferences, presentations and summary materials targeting consumers and their families will be developed. Such informal tailored presentations will involve our patients and their families in discussion of the study findings and implications of ongoing PROMs research for clinical practice.

### Access to study data

Deidentified data only will be made available through a nominated data repository on request. Public access to participant-level data will not be available due to patient privacy concerns.

### Publications policy

A publications policy has been created for this study. It outlines processes for authorship and will guide the appropriate recognition of all contributors to the intellectual property developed within the study. It applies to all publications that present research arising from and/or receiving funding from the pilot study as well as any future expanded studies based on the pilot findings.

**Author affiliations**
[1]NHMRC Clinical Trials Centre, The University of Sydney, Camperdown, New South Wales, Australia
[2]Faculty of Medicine and Health, The University of Sydney, Sydney, New South Wales, Australia
[3]Melanoma Institute Australia, Wollstonecraft, New South Wales, Australia
[4]Peter MacCallum Cancer Centre, Melbourne, Victoria, Australia

**Acknowledgements** We would like to acknowledge the significant contribution of other colleagues to the development of this protocol: Information technology staff (Ivy Tan and Ernie Wise); former project officers (Sam Robinson and Emma Zhang).

**Contributors** All authors participated in the drafting of this manuscript, undertook critical review of the intellectual content and approved the final version to be published. In addition, each of the authors made specific contributions as outlined below. KD was responsible for updating the original version of the protocol and drafting the first version of the manuscript, ensuring changes to the protocol were approved by the University of Sydney Human Research Ethics Committee and for coordinating coauthor responses. KD is also the corresponding author and guarantor of this manuscript and a member of the ePROMs-MEL Steering Committee. RM, RS, IB, SL and CL are Steering Committee members who were responsible for the conception and design of this protocol and study. RM is an experienced health services researcher and health economist, IB a clinical and research psychologist, SL an accomplished biostatistician and CL has extensive consumer representative experience in melanoma. All made substantial contributions to the conception and design of this research. FB contributed her expertise in the use of the psychometric measures, including the melanoma-specific Melanoma Concerns Questionnaire, and clinician–patient communication approaches for collected data. SD is an experienced clinical psychologist who

advised on the use of PROMs in the clinical setting and played a major role in the selection of appropriate PROMs. She is also the study psychologist who clinicians refer distressed study participants to for counselling. MD, an early career researcher and DM, a clinical nurse consultant, worked with the Steering Committee and second consumer representative LS, to review the PROMs and contributed to discussions on their appropriateness, particularly in terms of optimising research yield while minimising patient and clinic burden. JRT contributed his project officer skills and experience in psychosocial research and provided assistance to patients for data collection in the clinics when required. Melanoma clinicians TP, RS, AS, DS and JL-K contributed to patient recruitment and, along with FB, acted as clinical project champions for the delivery of psychological support services within these clinics.

**Funding** KD's salary is supported through the Australian National Health and Medical Research Council Centre for Research Excellence in Melanoma grant (Grant No.1135285). RPMS is supported by Melanoma Institute Australia. The project officer's salary is supported by a grant from The University of Sydney's DVCR Support Fund for COVID-19 impacted research. Support is also provided by Melanoma Institute Australia for IT programming and educational design, as well as access to the Melanoma Research Database to aid in the selection of eligible patients. The study funders had no role in, or influence on, the study design; collection, management, analysis and interpretation of data; writing of the report; or the decision to submit the report for publication.

**Competing interests** RS has received honoraria for advisory board participation from Merck Sharp & Dohme (MSD), Novartis and Qbiotics and speaking honoraria from Bristol Myers Squibb (BMS). All other authors declare no competing interests.

**Patient and public involvement** Patients and/or the public were involved in the design, or conduct, or reporting, or dissemination plans of this research. Refer to the Methods section for further details.

**Patient consent for publication** Not applicable.

**Provenance and peer review** Not commissioned; externally peer reviewed.

**ORCID iDs**
Kathy Dempsey http://orcid.org/0000-0003-4945-0857
Jake Robert Thompson http://orcid.org/0000-0002-0823-3218

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
