## [Reviewer comments · BMJ Open]

ARTICLE DETAILS

TITLE (PROVISIONAL)	Embedding electronic Patient Reported Outcome Measures into routine care for patients with Stage III MELanoma (ePROMs-MEL): protocol for a prospective, longitudinal, mixed-methods pilot study
AUTHORS	Dempsey, Kathy; Saw, Robyn; Bartula, Iris; Lo, Serigne; Lawn, Craig; Pennington, Thomas; Spillane, Andrew; Boyle, Frances; Dong, Skye; Dieng, Mbathio; Milne, Donna; Seaman, Linda; Saks, Dina; Lai-Kwon, Julia; Thompson, Jake; Morton, Rachael

VERSION 1 – REVIEW

REVIEWER	Fiona Kennedy University of Leeds, Psychosocial Oncology & Clinical Practice Research Group
REVIEW RETURNED	09-Aug-2022

GENERAL COMMENTS	Embedding electronic Patient Reported Outcome Measures into routine care for patients with Stage III MELanoma (ePROMs-MEL): Protocol of a prospective longitudinal Pilot study. A protocol paper which explores a prospective longitudinal pilot study to explore the use of ePROMS in melanoma follow-up care. The protocol could be strengthened with greater focus and clarity on the methodology in particular. Overall comments  1. Use of acronyms throughout is distracting, at times inconsistent and sometimes their first use is not spelt out in full (e.g. NHMRC – page 7) 2. It is not clear if this protocol/study is part of a wider programme of research. Latterly (page 16, line 29-30) the protocol refers to the 'main study' and it is unclear what this is referring to. 3. Need to outline in more detail the questionnaires used in terms of their description/format etc. 4. The overall methods section needs careful consideration in terms of how it is structured, use of subsections, repetition in places etc. 5. Authors should consult with the EQUATOR checklist (e.g. SPIRIT) for reporting protocols and ensure every appropriate item has been fully and clearly described. Title  6. I would advise that the authors carefully consider whether their study is a pilot or a feasibility study. See PLOS one paper Eldridge et al 2016 Abstract – Generally good clear introduction in the Abstract. Methods and analysis could be further refined in the following ways:  - Make clear the ePROMs completions are always done in
---

clinic/at hospital (i.e. not home completions); and all undertaken prior to clinic appointments? If a patient was late to appointment, would they still complete or not after appointment?

- Make clear the timescale of completions – I know these vary but is there a standard schedule that you could refer to?
- Make clear which ePROMs results (e.g. latter 3 questionnaires, plus DT/EQ5D or just latter 3 questionnaires) are presented in real-time?
- Give cut off in brackets for referral (this is given in the main text, so brief reference in Abstract would be useful)
- Give timeframe of qualitative interviews
- Give recruitment start dates – I appreciate this is an ongoing study, but reference to the start dates should be made in the Abstract and in the main text
- There is no mention in the Abstract of the surveys that both patients and staff complete (in addition to the interviews which are mentioned)

Introduction - This is generally good and provides a concise summary of contextual information.

7. Second paragraph – penultimate sentence – provide references of the mixed evidence you refer to

Methods - there are lots of clarifications needed in terms of the detail and presentation of the methods section.

8. Is it commonplace to provide explicit names of the consumer representatives? Has agreement been sought from the individuals concerned?

9. Page 7 – lines 18-22 – this is unclear. Need to explain more fully what clinician education resources were and how they were used – perhaps this needs it's own specific section?

10. Page 7 – lines 24-32 – this is unclear. How did the comprehensive database work? How did clinicians have access to it? Where was a stored? Did it receive the information automatically from the ipad software or was it manually entered? If so how regularly was it auto and/or manually updated?

11. Line stating “Dissemination of results to patients, clinicians and their general practitioners was strongly influenced by consumer views” – this is unclear and doesn't inform the reader of what dissemination was undertaken, how, when etc? Also is it relevant at this point in the protocol? I'm not sure if you are following a structure specified by the journal but having the PPI statement at the start of the methods seems to mean that some of the detail is repeated later.

12. Study design – the only reference to 18-months within the entire protocol and it is not clear what you are referring to in terms of an 18-months period? The recruitment period or were patients on the study for 18 months?

13. Reference to MIA and NSW – spell out if first time using acronym.

14. Sample size – no justification for 50 participants?

15. Recruitment – when did recruitment start? At each hospital? I would also outline any periods of study termination due to covid.

16. Patient recruitment – first sentence – repeats what already stated above.

17. Identification of participants – the MIA database is referred to – what about at RPAH? Are those patients also on the same database or was a different method used at this second hospital?

18. Be clear on what the study clinician reviewed in terms of clinical eligibility?

	19. Be clear on the process (Page 8, line 55) if a patient had not received the EoI? Assume they were resent it? Were they also telephoned again? 20. Page 9 – PIS acronym but not spelt out earlier and later referred to as PISCF. Be consistent in how you are referring to information sheets. 21. Page 9 – reference to RLM and RPMS – I wasn't sure what these were at first – e.g. is it an acronym used earlier/a place? On reflection now I think they are initials of the lead authors. If this is correct and even if not, make it more clear what you are referring to and perhaps just use two initials rather than 3-4 (if they are initials), especially as nowhere in the author list are 3-4 names spelt out, which led to my confusion. 22. Did the clinical nurse consultant always lead the baseline/clinic-based data collection? Rather than the project officer? 23. Page 9 – “Invitation to Participate”. Not sure it is necessary to capitalise. 24. Page 9 – clarify that only consenting clinicians saw study participants? Did all clinicians approached provide consent? 25. Interventions – ‘regularly’ – be more specific about the timescale of the study questionnaire completion. 26. MCQ-28 – spell out 27. PREMs – you do not outline anywhere what PREMs are (introduction?) – if you feel it is really relevant to state outline about PREMs you would need to define how different to PROMs for reader. 28. Questionnaires – I would expect to see more detail for each measure – e.g. number items, response options, scoring ranges, evidence of reliability/validity? 29. Page 10 – it is not clear how the results were presented to clinicians? From the current description of the number of questionnaires, there could be potentially a lot of information to present and without being explicit the reader is left wondering how this worked in practice? You could provide some screenshots illustrating how this appeared to the clinicians. Also refers to lines 28-32 of the same paragraph. 30. Similarly, it is not clear how results are presented to patients who request this – was this sent with a view to patients viewing it later (rather than before their imminent clinic appointment)? 31. Question scoring here could be outlined earlier (as per point 28 above) 32. Similar to above queries, how was a EORTC-QLQ-C30 change of 10+ points highlighted to clinicians? 33. Page 10, line 40-41 – “all rounds” – how many rounds? This needs to be clear throughout the protocol (e.g. Abstract etc). 34. Survey completed by patients, clinicians and clinic staff – clarify if all patients are being asked to complete this? Were these questionnaires done on ipads or was it a paper survey? 35. 10 patients and ‘some clinic staff’ – justify the 10 (why 10?) and ‘some’ seems very vague? In Figure 1 it is stated that 20% representative sample for the patient interviews – how did you ensure it was representative? Later you state purposively sampling using maximise response variation – but do not specify what criteria was followed in relation to maximising variation? 36. PISCF was used before, here it is spelt out in full again. 37. The “Interventions” section seems to be more than the interventions – it goes on to talk about the feedback/interviews/dissemination – these are not all ‘interventions’ so I think this section needs splitting.
--	--

	38. Figure 1 and Figure 2 – Could be much improved. A hierarchy flow chart might work better – e.g. see below, and also potentially a combined figure for both patients and clinicians. 39. Some of the detail/text in the flow chart has not been outlined in the main manuscript text. All key details of methods should appear in the text, and some detail in main text is not apparent in the figures. (e.g. regularly uploaded into redcap database – how/auto or manual/how often? Redcap isn't mentioned in the main text) 40. What does “Then 5 +/- 6 mean” in Figure 1? 41. Training for clinicians is outlined more in Figure 2 than it is in the main text. 42. Page 12, lines 38-42 – not clear how changes in baseline and follow-up scores will be explored? It is not clear how the level of change will be considered, and how this relates to feasibility/acceptability if it does at all? 43. Clinician perspectives – provide more detail on how these factors were presented/rated by clinicians? 44. Clinic staff perspectives – it is not clear if this is quantitative/qualitative – how feasibility/acceptability will be judged/assessed in this group? 45. Changes in referral rates, referral uptake, patients' QoL scores over time – how will this be assessed? Assume there will be some sort of comparison to earlier rates, but need to specify. And patient's QoL scores over time – assume this is within the patient's on study, rather than compare to earlier patient's QoL in the service, but needs clarifying. 46. Page 13 – ‘out of a maximum of 5’ – why a maximum of 5 times? This is not specified anywhere else. 47. Page 13 – final paragraph – ‘review resources prepared for this study’ – is this referring to the training that is referenced in Figure 2? If so this could be made much clearer – i.e. perhaps have a section about training clinicians at an appropriate place in the methods section? Also, did all clinicians watch these, were you able to monitor this? 48. Table 1 – T1, T2, T3 (end of study) – need more detail on the timing of these? It is stated that +/- 2 months but what is the expected norm? Also, relating to point 46 above – isn't there a maximum of 5 times, but here only baseline, T1, T2, T3 (4 time points)? 49. Page 14 – lines 43-44 – reference to Figure 3 isn't very helpful. If referring to this Figure, I think you need to be more specific about which of the outcomes specified in this Figure you are assessing as clearly some you are not assessing (E.g. costs, fidelity)? 50. Page 15 – line 5/6. How will you manage the variation of follow-up points? 51. Line 24/25 – ‘such as’ delete if using NVIVO 52. Qualitative analysis process needs further detail and clearer process. 53. Line 40/41 – ‘HREC’ spell out or state earlier 54. Line 47 – contradictory to what stated earlier about the first 2 questionnaires not being able to skip questions. 55. Page 16 – communication of trial results – this has mostly been stated before. 56. Unclear what is meant by “presentation and summary materials targeting consumers and their families will be developed”? 57. Publications policy refers to a ‘main study’ but this has not been referred to previously. If this is part of an overall grant or plans
--	--

	to gain funding to continue this work after the pilot/feasibility this needs to be outlined more fully earlier/perhaps even in the Introduction to set the context of this programme of research. Figures 58. See above points for comments about the Figures, which could be much improved to be more informative for the reader (points 37-40, and point 48)
--	--

REVIEWER	Cecilia Castillo Universidad de la República Facultad de Medicina
REVIEW RETURNED	15-Aug-2022

GENERAL COMMENTS	it's really good to see that this type of technology to support patients with a diagnosis of stage III melanoma stage III will be implemented taking in consideration that nowadays standard adjuvant treatments are long and could probably affect frequently their well being and quality of life In order to be able to reproduce the study, I would like to see the questionnaires they will perform to evaluate acceptability of patients and doctors
--

VERSION 1 – AUTHOR RESPONSE

Reviewer: 1

Dr. Fiona Kennedy, University of Leeds Comments to the Author:

Overall comments

1. Use of acronyms throughout is distracting, at times inconsistent and sometimes their first use is not spelt out in full (e.g. NHMRC – page 7)

Thank you for highlighting this distraction. We have removed the acronyms where possible. Where this is not practicable, we have made sure that all are spelt out in full on first use. The following terms remain as acronyms as we believe they are widely known and used: PROMS, ePROMs, questionnaire names, PIS, PISCF and Eol. We have also retained the acronyms for the two sites the study is being conducted at (MIA and RPAH) as these are mentioned quite frequently.

2. It is not clear if this protocol/study is part of a wider programme of research. Latterly (page 16, line 29-30) the protocol refers to the 'main study' and it is unclear what this is referring to.

This comment was referring specifically to the publications policy. It was meant to convey that this policy related to not only publications from the current pilot study, but also to any other expanded studies that may stem from this pilot study. This sentence has now been reworded to make this point clearer.

"It applies to all publications that present research arising from and/or receiving funding from the pilot study as well as any future expanded studies based on the pilot findings."

3. Need to outline in more detail the questionnaires used in terms of their description/format etc.

Word limitations restrict our ability to provide more details on each questionnaire. We have added Appendix 1, which contains the five validated questionnaires used in the study. All except the Melanoma Concerns Questionnaire are commonly used in psychosocial research. Interested readers can find out more details from the references in this publication.

4. The overall methods section needs careful consideration in terms of how it is structured, use of subsections, repetition in places etc.

We have made some changes to the structure of this section as suggested by the editor. Examples of repetition highlighted in your comments have been removed.

5. Authors should consult with the EQUATOR checklist (e.g. SPIRIT) for reporting protocols and ensure every appropriate item has been fully and clearly described.

Our Protocol document closely follows the SPIRIT guidelines. The requirements for publishing information about the Protocol varies from this SPIRIT checklist and we have used the BMJO guidelines in the preparation of this manuscript.

6. Title I would advise that the authors carefully consider whether their study is a pilot or a feasibility study. See PLOS one paper Eldridge et al 2016.

Thank you for the reference to this interesting paper. As Eldridge et al. acknowledge, 'feasibility' and 'pilot' studies are not mutually exclusive, and their own framework considered pilot studies as a subset of feasibility studies. Both "ask whether something can be done, should we proceed with it, and if so, how." A pilot study acts as a sample, smaller scale study for a subsequent larger future study. This is exactly what our study does. Hence, we will continue to use the term 'pilot' study.

Abstract

Generally good clear introduction in the Abstract. Methods and analysis could be further refined in the following ways:

- Make clear the ePROMs completions are always done in clinic/at hospital (i.e. not home completions); and all undertaken prior to clinic appointments? If a patient was late to appointment, would they still complete or not after appointment?
- Make clear the timescale of completions – I know these vary but is there a standard schedule that you could refer to?
- Make clear which ePROMs results (e.g. latter 3 questionnaires, plus DT/EQ5D or just latter 3 questionnaires) are presented in real-time?
- Give cut off in brackets for referral (this is given in the main text, so brief reference in Abstract would be useful)
- Give timeframe of qualitative interviews
- Give recruitment start dates – I appreciate this is an ongoing study, but reference to the start dates should be made in the Abstract and in the main text
- There is no mention in the Abstract of the surveys that both patients and staff complete (in addition to the interviews which are mentioned)

We have done our best to briefly address all of the above points within the abstract's 300 word limit.

Introduction

This is generally good and provides a concise summary of contextual information.

7. Second paragraph – penultimate sentence – provide references of the mixed evidence you refer to

Done (same reference – 9).

Methods

There are lots of clarifications needed in terms of the detail and presentation of the methods section.

8. Is it commonplace to provide explicit names of the consumer representatives? Has agreement been sought from the individuals concerned?

Thank you for identifying this as a possible concern. We have removed this sentence from the manuscript.

9. Page 7 – lines 18-22 – this is unclear. Need to explain more fully what clinician education resources were and how they were used – perhaps this needs its own specific section?

This section is just in relation to consumer involvement. Further information on these resources is provided in the last paragraph of the secondary outcomes section.

10. Page 7 – lines 24-32 – this is unclear. How did the comprehensive database work? How did clinicians have access to it? Where was it stored? Did it receive the information automatically from the iPad software or was it manually entered? If so how regularly was it auto and/or manually updated?

This excerpt is again part of the 'Patient and public involvement' section, which outlines the role of consumers in the study. It is not meant to provide a detailed description of the intervention, which is provided in the 'Interventions' section.

The first sentence of the fourth paragraph of the 'Interventions' section has been amended as follows:

"Results from the completed questionnaires will be automatically generated, emailed to the clinic secretary and printed for the treating clinician in real-time, so the results can be discussed during their consultation, along with medical concerns."

11. Line stating "Dissemination of results to patients, clinicians and their general practitioners was strongly influenced by consumer views" – this is unclear and doesn't inform the reader of what dissemination was undertaken, how, when etc? Also is it relevant at this point in the protocol? I'm not sure if you are following a structure specified by the journal but having the PPI statement at the start of the methods seems to mean that some of the detail is repeated later.

The sentence you refer to has been reworded:

"Consumer views also strongly influenced the plan for dissemination of results to patients, clinicians and their general practitioners".

The PPI statement is a journal formatting requirement with its position specified as being at the beginning of the 'Methods and analysis' section. The dissemination plan is mentioned here as another example of consumer involvement in the study. Further information about dissemination of results is included in the relevant 'Ethics and dissemination' section.

12. Study design – the only reference to 18-months within the entire protocol and it is not clear what you are referring to in terms of an 18-months period? The recruitment period or were patients on the study for 18 months?

Thank you for pointing out this lack of clarity. The sentence "This pilot study is based on a single cohort of patients over an 18-month period (including periods of inactivity due to COVID-19)." has

been removed from the first paragraph of the 'Study design' section and replaced with the following sentence in the 'Recruitment' section:

"Thirty-six patients and seven clinicians have been recruited since the study commenced in May 2021, with the study expected to be completed by June 2023."

We have also added the following sentence to the first paragraph of the 'Interventions' section.

"Questionnaires will be administered at baseline, with a minimum of two and a maximum of three, follow-up time points."

13. Reference to MIA and NSW – spell out if first time using acronym.

Done.

14. Sample size – no justification for 50 participants?

Justification for the reduction in sample size has been added to the Eligibility criteria and sample size section:

"This research project has experienced substantial delays to recruitment due to research staff being unable to enter clinic sites during COVID-19. In April 2022, the decision was made to reduce the initial recruitment target of 100 to 50, so that the trial may be completed before external funding expires. The target reduction is justified on the basis that this project is a non-randomised feasibility pilot study. Assuming 30% of Stage III melanoma patients have levels of distress that indicate the need for clinical intervention,¹³ we would expect to identify approximately 17 patients within our sample who will be triaged for discussion of referrals to support services. This number is considered sufficient to inform a decision on whether or not to proceed to a larger-scale trial with additional sites."

15. Recruitment – when did recruitment start? At each hospital? I would also outline any periods of study termination due to covid.

The following information has been added to the 'Recruitment' section:

"Recruitment has been intermittent due to delays caused by COVID-19 (April-September 2020 and June 2021-March 2022). Recruitment commenced at MIA on 28 May 2021 and at RPAH on 5 July 2021."

16. Patient recruitment – first sentence – repeats what already stated above.

That sentence has been deleted.

17. Identification of participants – the MIA database is referred to – what about at RPAH? Are those patients also on the same database or was a different method used at this second hospital?

Recruited patients at both MIA and RPAH are seen by MIA clinicians and their details are all entered on the MIA database.

18. Be clear on what the study clinician reviewed in terms of clinical eligibility?

A patient's clinical eligibility may have changed since the data was entered into the MIA database. The following has been added to the Patient recruitment process section.

"(i.e. are still under the care of MIA clinicians at either site or have not progressed to Stage 4 melanoma)"

19. Be clear on the process (Page 8, line 55) if a patient had not received the Eol? Assume they were resent it? Were they also telephoned again?

This information is already contained in the second paragraph of the Patient recruitment process section:

“If patients do not respond to the request to return the Eol, the project officer will telephone them once to check whether they have received it.”

20. Page 9 – PIS acronym but not spelt out earlier and later referred to as PISCF. Be consistent in how you are referring to information sheets.

This has now been corrected and is consistent throughout the manuscript.

21. Page 9 – reference to RLM and RPMS – I wasn't sure what these were at first – e.g. is it an acronym used earlier/a place? On reflection now I think they are initials of the lead authors. If this is correct and even if not, make it more clear what you are referring to and perhaps just use two initials rather than 3-4 (if they are initials), especially as nowhere in the author list are 3-4 names spelt out, which led to my confusion.

These initials have been removed from the text.

22. Did the clinical nurse consultant always lead the baseline/clinic-based data collection? Rather than the project officer?

The secretary may also send the list of eligible patients to the project officer, who, for patient confidentiality reasons, was not given access to the patient database. That sentence has been amended as follows:

The clinical nurse consultant or clinic secretary will then send the approved list of eligible patients to the project officer.

23. Page 9 – “Invitation to Participate”. Not sure it is necessary to capitalise.

This term is no longer capitalised.

24. Page 9 – clarify that only consenting clinicians saw study participants? Did all clinicians approached provide consent?

Only consenting clinicians saw the study participants in terms of reviewing the study questionnaires with them. Other clinicians may see the study participants as part of their routine care. All clinicians who were approached provided consent.

The following sentence was added as the last sentence of the ‘Clinician recruitment process’ section:

All clinicians who were approached to participate consented to the study and reviewed the study questionnaires with the patients.

25. Interventions – ‘regularly’ – be more specific about the timescale of the study questionnaire completion.

This issue has been addressed in point 12 above.

26. MCQ-28 – spell out

Done

27. PREMs – you do not outline anywhere what PREMs are (introduction?) – if you feel it is really relevant to state outline about PREMs you would need to define how different to PROMs for reader.

Thank you for picking this up. We have deleted the comments about PREMs.

28. Questionnaires – I would expect to see more detail for each measure – e.g. number items, response options, scoring ranges, evidence of reliability/validity?

See our response to point 3. We have added the following sentence to the end of the second paragraph in the 'Interventions' section.

“All questionnaires are provided in Appendix 1.”

29. Page 10 – it is not clear how the results were presented to clinicians? From the current description of the number of questionnaires, there could be potentially a lot of information to present and without being explicit the reader is left wondering how this worked in practice? You could provide some screenshots illustrating how this appeared to the clinicians. Also refers to lines 28-32 of the same paragraph.

The results are printed out for the clinician to review with the patient. We have added an Appendix 2 which contains further information on the reporting processes including images of the report formats.

Clinicians receive the overall QoL score and overall health score from this questionnaire and study reports from each visit are scanned into their patient records. For this pilot study, clinicians will refer back to previous scores to assess change. We agree data visualisations of temporal scores showing change over time would be ideal, and we plan to provide clinicians with these graphs in a future expanded study.

30. Similarly, it is not clear how results are presented to patients who request this – was this sent with a view to patients viewing it later (rather than before their imminent clinic appointment)?

Patients who request a copy of their questionnaire report will be receive the same email as the clinician at the same time. They may view their report on their tablet device prior to their appointment, but printed copies are not provided to the patients. They may also choose to view it at a later time. Further information about this process will be included in Appendix 2.

31. Question scoring here could be outlined earlier (as per point 28 above)

See our response to point 3.

32. Similar to above queries, how was a EORTC-QLQ-C30 change of 10+ points highlighted to clinicians?

Clinicians receive the overall QoL score and overall health score from this questionnaire and study reports from each visit are scanned into their patient records, so the clinicians can check previous scores for their patients who have low QoL scores on other measures. If this pilot study is subsequently expanded, tracking of individual questionnaire scores will be automated as part of the reporting process. Further information is available in Appendix 2.

33. Page 10, line 40-41 – “all rounds” – how many rounds? This needs to be clear throughout the protocol (e.g. Abstract etc).

The following words have been added to that sentence for clarification and an abbreviated version has also been added to the abstract.

“After all rounds of patient questionnaires (baseline plus 2-3 follow-up rounds per patient) have been completed ...”

34. Survey completed by patients, clinicians and clinic staff – clarify if all patients are being asked to complete this? Were these questionnaires done on ipads or was it a paper survey?

The following amendments have been made to that sentence:

“... all patients, clinicians and clinic staff involved in the project will be asked to complete an electronic survey providing quantitative and open-ended assessments of their experiences with the PROMs questionnaires.

35. 10 patients and ‘some clinic staff’ – justify the 10 (why 10?) and ‘some’ seems very vague?

The following amendments have been made to that sentence:

“In addition, approximately 20% of patients (10 patients), all study clinicians and clinic staff with major involvement in the study will be asked ...”

In Figure 1 it is stated that 20% representative sample for the patient interviews – how did you ensure it was representative? Later you state purposively sampling using maximise response variation – but do not specify what criteria was followed in relation to maximising variation?

Thank you for picking up this discrepancy in sampling. The relevant text in Figure 1 has been changed to read “20% of participants ...”

The text has been amended to read as follows:

“Patients will be purposively sampled to maximise response variation, based on a range of individual views and experiences as identified from their survey results.”

36. PISCF was used before, here it is spelt out in full again.

This has now been corrected.

37. The “Interventions” section seems to be more than the interventions – it goes on to talk about the feedback/interviews/dissemination – these are not all ‘interventions’ so I think this section needs splitting.

Thank you for noting this. Sections on inclusion of patients who have progressed to Stage IV and dissemination have been removed from this section. We believe the reporting, survey and interview components are all part of the intervention processes.

38. Figure 1 and Figure 2 – Could be much improved. A hierarchy flow chart might work better – e.g. see below, and also potentially a combined figure for both patients and clinicians.

We appreciate your feedback on the figures, but believe Figures 1 and 2 are fit for purpose.

39. Some of the detail/text in the flow chart has not been outlined in the main manuscript text. All key details of methods should appear in the text, and some detail in main text is not apparent in the figures. (e.g. regularly uploaded into redcap database – how/auto or manual/how often? Redcap isn’t mentioned in the main text)

We believe that figures and tables are meant to provide additional information to the reader and not to duplicate information provided in the text. Information concerning the uploading of data into the REDCap database, which does not form part of the intervention, will be mentioned in Appendix 2.

40. What does “Then 5 +/- 6 mean” in Figure 1?

This refers to Step 5 with or without Step 6, depending on the score from the Distress Thermometer. We realise now that this is confusing and have deleted it from Figure 1.

41. Training for clinicians is outlined more in Figure 2 than it is in the main text.

We believe this is an appropriate place for this information. Figure 2 is referenced in the text to avoid duplication of information.

42. Page 12, lines 38-42 – not clear how changes in baseline and follow-up scores will be explored? It is not clear how the level of change will be considered, and how this relates to feasibility/acceptability if it does at all?

We believe this is covered in the 'Data analysis/Statistical methods' section. Further information will be provided in Appendix 2.

43. Clinician perspectives – provide more detail on how these factors were presented/rated by clinicians?

The clinician perspectives will be assessed in the end-of-study surveys and interviews. The end-of-study surveys for patients, clinicians and clinic staff are contained in the new Appendix 3.

44. Clinic staff perspectives – it is not clear if this is quantitative/qualitative – how feasibility/acceptability will be judged/assessed in this group?

As above, they will be assessed through quantitative surveys (see Appendix 3) and a subset of clinic staff will also undergo qualitative interviews.

45. Changes in referral rates, referral uptake, patients' QoL scores over time – how will this be assessed? Assume there will be some sort of comparison to earlier rates, but need to specify. And patient's QoL scores over time – assume this is within the patient's on study, rather than compare to earlier patient's QoL in the service, but needs clarifying.

This information on assessment of referral rates, uptake and QoL scores over time is covered in the 'Data analysis / Statistical methods' section. It will be largely based on descriptive statistics. As noted in the first paragraph of the 'Study design and setting' section, "Although there is no control group, the design incorporates before-and-after study design elements allowing for comparisons of psychosocial health and quality of life in individuals with Stage III melanoma before and after the introduction of ePROMs.

46. Page 13 – 'out of a maximum of 5' – why a maximum of 5 times? This is not specified anywhere else.

Thank you for picking up this error. This refers to the maximum number of times each patient will complete the questionnaires. Before the COVID delays, we had thought we may be able to obtain data at baseline and four other time points (making a maximum of 5), but this had to be reduced to four (baseline plus 2 or 3 follow-up appointments). This has now been corrected in the text.

47. Page 13 – final paragraph – 'review resources prepared for this study' – is this referring to the training that is referenced in Figure 2? If so this could be made much clearer – i.e. perhaps have a section about training clinicians at an appropriate place in the methods section? Also, did all clinicians watch these, were you able to monitor this?

Yes, this does refer to the training materials. This is a secondary outcome that will be discussed with clinicians and assessed at the end-of-study interviews.

48. Table 1 – T1, T2, T3 (end of study) – need more detail on the timing of these? It is stated that +/- 2 months but what is the expected norm? Also, relating to point 46 above – isn't there a maximum of 5 times, but here only baseline, T1, T2, T3 (4 time points)?

As noted in our response to point 46, four time points is correct. The schedule of patient appointments dictates the options for questionnaire completion. Appointments range from 3 weekly to 6 monthly and patients are not always able to come in 30 minutes beforehand to complete the questionnaires.

49. Page 14 – lines 43-44 – reference to Figure 3 isn't very helpful. If referring to this Figure, I think you need to be more specific about which of the outcomes specified in this Figure you are assessing as clearly some you are not assessing (E.g. costs, fidelity)?

Given the word limitations, we have not provided details about the implementation metrics for this current publication. When publishing our findings, we will aim to do assessments on as many of the implementation outcomes as possible, including costs (e.g. clinic staff time) and fidelity to the protocol (e.g. changes required to manage COVID delays).

50. Page 15 – line 5/6. How will you manage the variation of follow-up points?

*We will use linear mixed effects modelling to account for repeated measures and differential time points among participants. This is similar to analysis of longitudinal data from a clinical trial where measurements at follow up time points are missing. See De Livera AM, Zaloumis S, Simpson JA. Models for the analysis of repeated continuous outcome measures in clinical trials. *Respirology*,2014;19:155–161 doi: 10.1111/resp.12217*

51. Line 24/25 – 'such as' delete if using NVIVO

Done.

52. Qualitative analysis process needs further detail and clearer process.

We think what we have described is sufficient – the qualitative analysis process is iterative so it is not possible to definitively describe in advance.

53. Line 40/41 – 'HREC' spell out or state earlier

Corrected.

54. Line 47 – contradictory to what stated earlier about the first 2 questionnaires not being able to skip questions.

Is correct as is – states that all questions in first 2 questionnaires are mandatory.

55. Page 16 – communication of trial results – this has mostly been stated before.

The duplication was previously from the 'Innovations' section; it has now been removed from there.

56. Unclear what is meant by "presentation and summary materials targeting consumers and their families will be developed"?

What we mean by this is that we will tailor presentations of our results to patients and families and invite them to come to MIA to review and discuss implications of the findings for changes to clinic practice. The following sentence has been added to the end of that paragraph:

"Such informal tailored presentations will involve our patients and their families in discussion of the study findings and implications of ongoing PROMs research for clinical practice."

57. Publications policy refers to a 'main study' but this has not been referred to previously. If this is part of an overall grant or plans to gain funding to continue this work after the pilot/feasibility this needs to be outlined more fully earlier/perhaps even in the Introduction to set the context of this programme of research.

This issue was addressed in our response to point 2.

Figures

58. See above points for comments about the Figures, which could be much improved to be more informative for the reader (points 37-40, and point 48)

Our responses to these comments are noted in the previous points.

Reviewer: 2

Cecilia Castillo, Universidad de la República Facultad de Medicina Comments to the Author:

It's really good to see that this type of technology to support patients with a diagnosis of stage III melanoma stage III will be implemented taking in consideration that nowadays standard adjuvant treatments are long and could probably affect frequently their well being and quality of life.

In order to be able to reproduce the study, I would like to see the questionnaires they will perform to evaluate acceptability of patients and doctors.

The three end-of-study questionnaires (for patients, clinicians and clinic staff) have been combined in Appendix 2 titled.

VERSION 2 – REVIEW

REVIEWER	Fiona Kennedy University of Leeds, Psychosocial Oncology & Clinical Practice Research Group
REVIEW RETURNED	11-Oct-2022

GENERAL COMMENTS	Many thanks for the authors for addressing previous queries and submitting this revised version. Most of my previous comments have been addressed, but i have a few outstanding queries related to the earlier comments and the authors responses: 1) As the editor stated, the strengths and limitations section/bullet points currently only refers to strengths, and does not include any limitations, which i think the editor wished to be revised/included.2) I would suggest referring to Appendix 3 earlier in the Outcomes section3) In response to point 50, you state you will use linear mixed effects modelling, but this is not described at all during the statistical analysis section within the manuscript.4) I appreciate the difficulties of word limits in relation to point 49. However, I think rather than present the Proctor framework in the manuscript (Figure 3) which can still be referenced, it would make much more sense and be more insightful for the reader if you gave more detail (in a Table/Figure, so not increasing words) of what the plans for assessing implementation outcomes were (including proposed outcomes such as staff time, changes required etc). This doesn't need to provide exact details of assessment but giving some indication of the outcomes would be valuable. In any subsequent results paper you can then refer back to what you said in the protocol paper and provide detail and results.5) Qualitative analysis - I appreciate your response to point 52, but it would be commonplace to be upfront about your qualitative methodology/approach, for example whether you are taking an inductive or deductive approach. It sounds from what you are saying
---

	that it will partially include quantification of what patients say (e.g. whether results were discussed). More explicit detail on the approach and proposed analysis should be included. 6) In relation to point 54, I am still confused about whether questions were mandatory or able to be skipped, and the ethics section seems to contradict what is stated in the main text. The main text is clear in stating that the first two triage questionnaires all questions were mandatory, but in the ethics section it implies that all questions can be skipped. I think this could be clarified to make it clear to the reader.
--	---

VERSION 2 – AUTHOR RESPONSE

Reviewer: 1

Dr. Fiona Kennedy, University of Leeds

Comments to the Author:

Many thanks for the authors for addressing previous queries and submitting this revised version. Most of my previous comments have been addressed, but I have a few outstanding queries related to the earlier comments and the authors responses:

1) As the editor stated, the strengths and limitations section/bullet points currently only refers to strengths, and does not include any limitations, which I think the editor wished to be revised/included.

Thank you for highlighting this misunderstanding – it has now been corrected (see above).

2) I would suggest referring to Appendix 3 earlier in the Outcomes section

We have now divided Appendix 3 into 3a, 3b and 3c and have mentioned them in each of the relevant parts of the Outcomes section.

3) In response to point 50, you state you will use linear mixed effects modelling, but this is not described at all during the statistical analysis section within the manuscript.

Thank you for picking up this oversight – we added this information in our previous response to you, but did not add it to the text. This omission has now been corrected (see the statistical analysis section of the manuscript) and we have added this additional reference to the references list.

4) I appreciate the difficulties of word limits in relation to point 49. However, I think rather than present the Proctor framework in the manuscript (Figure 3) which can still be referenced, it would make much more sense and be more insightful for the reader if you gave more detail (in a Table/Figure, so not increasing words) of what the plans for assessing implementation outcomes were (including proposed outcomes such as staff time, changes required etc). This doesn't need to provide exact details of assessment but giving some indication of the outcomes would be valuable. In any subsequent results paper you can then refer back to what you said in the protocol paper and provide detail and results.

We have replaced the original Table 2 (from Proctor et al.) with a new Table 2 which defines the additional implementation outcomes (excluding acceptability and feasibility which are described throughout the protocol) and provides examples of how they will be measured in the context of this study.

5) Qualitative analysis - I appreciate your response to point 52, but it would be commonplace to be upfront about your qualitative methodology/approach, for example whether you are taking an inductive or deductive approach. It sounds from what you are saying that it will partially include quantification of what patients say (e.g. whether results were discussed). More explicit detail on the approach and proposed analysis should be included.

We have added the following sentence to the beginning of this section:

This study takes an inductive approach to data analysis, with no pre-determined hypotheses about what we expect to find.

The only qualitative component of this study is the interviews. As is usual in qualitative research practice, we will not be analysing this data in a quantitative manner (for example we will not be counting how many people said X and putting that in a table). To make this point as clear as possible, we have changed the wording of the following sentence.

Electronic qualitative analysis software (NVivo) will be used to assist in the creation ~~development~~ of a framework of key themes identified from the interview data that will then inform the development of an iterative thematic analysis process.

6) In relation to point 54, I am still confused about whether questions were mandatory or able to be skipped, and the ethics section seems to contradict what is stated in the main text. The main text is clear in stating that the first two triage questionnaires all questions were mandatory, but in the ethics section it implies that all questions can be skipped. I think this could be clarified to make it clear to the reader.

Apologies for this oversight. The wording in the ethics section has now been changed to read:

If a patient does not wish to answer a question, they may skip it and go to the next question (if they are completing the second set of questionnaires), or immediately stop completing the questionnaires (if they are completing the first set of questionnaires, where all questions are mandatory).

VERSION 3 – REVIEW

REVIEWER	Fiona Kennedy University of Leeds, Psychosocial Oncology & Clinical Practice Research Group
REVIEW RETURNED	05-Dec-2022
GENERAL COMMENTS	Thank you for addressing all my previous comments - the manuscript is much clearer now.